# Plant Biodiversity Knowledge Varies by Gender in Sustainable Amazonian Agricultural Systems Called Chacras

**Carmen X. Luzuriaga-Quichimbo [1], Míriam Hernández del Barco [2], José Blanco-Salas [3],\*, Carlos E. Cerón-Martínez [4] and Trinidad Ruiz-Téllez [3]**

[1]   CENBIO, Universidad UTE, Quito 170147, Ecuador
[2]   Departamento de Didáctica de las Ciencias Experimentales y Matemáticas, Facultad de Educación, Universidad de Extremadura, 06071 Badajoz, Spain
[3]   Departmento de Biología Vegetal, Ecología y Ciencias de la Tierra, Universidad de Extremadura, 06006 Badajoz, Spain
[4]   Herbario Alfredo Paredes, QAP, Universidad Central de Ecuador, Quito 170147, Ecuador
\*   Correspondence: blanco_salas@unex.es; Tel.: +34-924-289-300

**Abstract:** Chacras, which are Amazonian agricultural systems, are examples of traditional agricultural management that are sustainable. They are also characteristic of the identities of different ethnographic groups in tropical America. However, information regarding the botanical characterization of chacras is scant. In tropical rural communities, there is a gender bias hypothesis that makes women potential reservoirs of traditional chacras plant knowledge. We present an experimental study in order to demonstrate if this knowledge difference really exists and to plan accordingly. We performed workshops in an isolated Kichwa community from Amazonian Ecuador. We calculated the cultural signififcance index (CSI) for 97 local flora plants. Our results revealed statistically significant differences. They were coherent with the Kichwa worldview and the structure of their society. We concluded that gender perspective must be taken into account in biodiversity conservation programs, such as, for example, those to implement the resilient agricultural practices of tropical contexts promoted by The United Nations Sustainable Development Goal 2 (SGD2).

**Keywords:** biodiversity conservation; Amazonian indigenous; women; agroecological production; livelihood; economic growth; sustainable development goals; security food; traditional knowledge; ethnobotany

## 1. Introduction

Agriculture provides livelihoods for 40% of the global population and is the largest source of income for poor rural households. The United Nations Sustainable Development Goal 2 (SGD2) has a target to double the agricultural productivity and incomes of indigenous people, women, and other small-scale food producers [1]. In developing countries, women make up about 43% of the agricultural labor force.

The challenge is clear in tropical America [2], and uniquely in the 6.5 million km$^2$ occupied by the Amazon basin for which development and conservation are in the spotlight of controversial debates [3,4]. Within it, the Ecuadorian Amazon forest (EAF) is considered a salient biodiversity hotspot. In recent decades, it has been seriously affected by rapid changes in land use, oil extraction, and mining [5] that has impacted sustainable development [6,7]. Despite its biological richness, a large number of inhabitants of this area still suffer from food supply insecurity (up to 40%; 25% in Ecuador) [8]. Global food security has to be taken into account not only on a large scale but also

on any significant smaller scale. Amazonian populations must be considered because of their social, ethnobiological, and environmental importance. The Convention on Biological Diversity recognized that the use of technologies that restrict genetic management in crop species could adversely affect traditional practices of seed exchange, infringe cultural values, and increase gaps between institutions and local cultures [9]. In the international Declaration of Atitlan-Guatemala 2002, the collective right to food sovereignty was declared as essential for the continuation of indigenous identities [10]. As indigenous women are the most important actors in the assurance of food for their families and communities, at the IV Continental Meeting of Indigenous Women of the Americas-in Peru (2004) [11], they expressed their concern about the implementation of megaprojects that could not benefit their territories, knowledge, and natural resources. They ratified their commitment to retake their ancestral foods, wit, knowledge, and wisdom [11].

For these reasons, we aimed to quantify the importance of this female knowledge in biodiversity conservation, focusing on [12] the best sustainable system of EAF agriculture: the chacras. We would like to emphasize the importance of connecting these management practices with food sovereignty.

The chacra is the migratory system of cultivation developed in the Amazonian highlands, opposite to the one semi-intensively developed in the floodplains [13]. It is based on the practice of slash-and-burn: older trees are cut, minor vegetation is cleaned and finally burned, producing ashes, so minerals are easy to be assimilated. A short period of use on a plot is followed by an extended period of abandonment. The abandoned plot continues producing different varieties of plants also used by the inhabitants, and, finally, the Ecuadorian Amazonian tropical forest regenerates, as summarized in Figure 1.

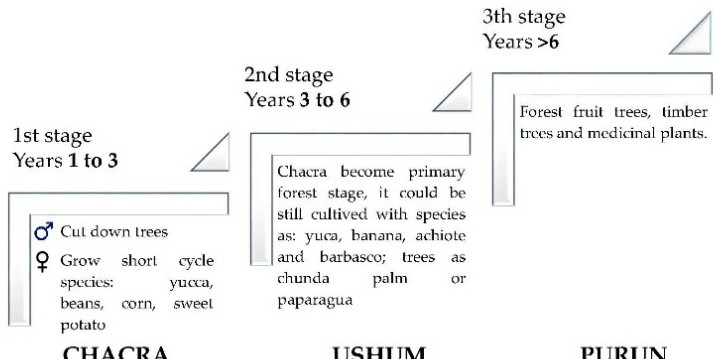

**Figure 1.** Summary of the migratory system of cultivation developed in the Amazonian tropical forest (Source: personal collection).

The stratified plant architecture and foliage density control solar radiation, temperature, humidity, and weeds, reducing soil erosion under torrential rain, and the mixed farming or polyculture reduces the multiplication of pests and diseases. Although chacras are situated on very acidic and infertile soils, basically constituted of clay and sand, they have enabled the autochthonous Amazonian civilizations to subsist, thanks to the management of many species and varieties that have provided very diversified diets [14]. The traditional management of chacras is an example of sustainable Amazonian agriculture. It is also a characteristic of the identity of each ethnographic group [15].

Although there is an extensive ethnographical bibliography on Ecuadorian Amazonian indigenous populations, information regarding the botanical characterization of chacras is scant [16]. In addition to this, in tropical rural communities, ethnobotanical knowledge has been suggested as gender-specific [17–19]. Men should specialize in knowledge about the forests, whereas women tend to be more informed about chacras and the disturbance of species associated with human habitats. This gender bias hypothesis makes women potential reservoirs of the traditional biodiversity of chacras plant knowledge.

The approach of specialists introducing the gender perspective in ethnobotanical studies of diverse places of the world is contributing a new vision to the Programs of Conservation of the Vegetal Biodiversity [20–22].

The null hypothesis in our study is that women and men sustain roughly equal knowledge of the local biodiversity of chacras. We designed fieldwork in a remote Amazonian community, which has received little outside contact. A very efficient methodology was set up. It was adapted to the climatic, sociocultural, and economic conditions. The experience can be replicated in other difficult locations. It enables easy reliable diagnosis that can also be uploaded into larger databases.

## 2. Materials and Methods

### 2.1. Study Area and Permissions

We selected an isolated community characteristic of the Amazonian ancestral societies in Pastaza (Ecuador). The Kichwa community of Pakayaku (01°39′07″ S–77°36′11″ W) is accessible only after 5 h of navigation along the Bobonaza River, without connection by plane or road traffic [23]. Its original territory comprises 40,985 ha [24] is located in the wettest and rainiest region of the country (>5400 L/m$^2$), with an average annual temperature of 25 °C [25]. It is formed by about 1000 people that live scattered in the forest in isolated cabins of wood and straw, integrated with the surrounding environment. There is no mobile telephony, internet, electricity, running water, sewerage, nor sanitation, and the radio is just used for emergencies [24,26]. There is a public school, but more than 40% of the population is illiterate. Pakayaku access is not easy, because you cannot get into the territory if you are not allowed by the kuraka, President, and traditional Kichwa authorities. A council has the capacity to decide on the basis of the decisions taken in the Assembly [27]. In 2015, the indigenous authorities chaired by the President Luzmila Gayas met in a specific Assembly. They gave a one-year authorization for researcher C.X.L.-Q. to carry out a workshop and to catalog useful plants of Pakayaku. It was also necessary to request collection and transfer permits from the Ministry of Environment in accordance with the Legislation of Ecuador, following the regulations of the Nagoya Protocol of the United Nations Convention on Biological Diversity [28]. The reference permission signature is MAE-DPAP-2016-2243.

### 2.2. Plant Material

Assisted by the Assembly of the community, who estimated there are a total number of 100 chacras in Pakayaku, 10% were selected to represent the variability of the territory (Figure 2). They were the following:

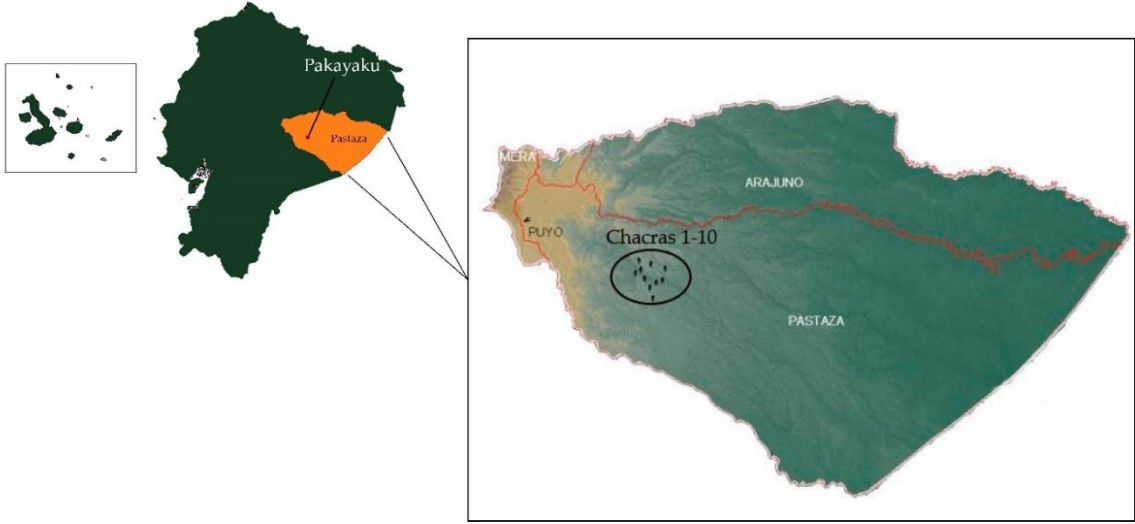

**Figure 2.** Location of the studied chacras in Pastaza Province, Ecuador.

*Chacra 1* Mr. Ramón Aranda, half an hour's walk from the community. 01°39′07,2″ S–077°36′11,8″ W. 380 m.a.s.l. *Chacra 2* Mrs. Venika Aranda, half an hour's walk from the community. 01°38′56″ S–077°36′34″ W. 406 m.a.s.l. *Chacra 3* Mrs. Ana Aranda, half an hour's walk from the community 01°38′55″ S–077°36′31″ W. 403 m.a.s.l. *Chacra 4* Mrs. Alicia Tapuy, a forty minute walk from the community. 01°40′51″ S–077°35′24″ W.

416 m.a.s.l. *Chacra 5* Mrs. Erika Gayas, half an hour's walk from the community. 01°39′44″ S–077°35′40″ W. 370 m.a.s.l. *Chacra 6* Mr. Aparicio Aranda, a two-hour walk from the community. 01°39′03″ S–077°34′36″ W. 592 m.a.s.l. *Chacra 7* Mr. Olga Aranda. 01°38′25″ S–077°36′23″ W. 378 m.a.s.l. *Chacra 8* Mr. Luis Santi, half an hour's walk from the community. 01°38′45″ S–077°35′40″ W. 43 m.a.s.l. *Chacra 9* Mrs. Julia Mayancha, a forty minute walk from the community. 01°39′39,6″ S–077°35′31,2″ W. 382 m.a.s.l. *Chacra 10* Mr. Octavio Aranda, a one hour walk from the community. 01°39′50,4″ S–077°35′49,2″ W. 417 m.a.s.l.

Sampling was made by transects of 50 × 2 m following Cerón (2003). Plants were photographed, collected, and preserved in plastic bags with ethanol 70% for transport by canoe and afterward by road to Quito (Herbario Alfredo Paredes QAP, Universidad Central de Ecuador). They were treated with the standard herbarium protocols (Ceron 2015) for desiccation and mounting. Taxonomical identification was made in the context of the preparation of the Doctoral Ph. Master Thesis of C.X.L.-Q. (Luzuriaga, 2017) at the QAP Herbarium and supervised by C.E. C-M. Families, and upper taxa were ordered following the APG IV System (2016). Accepted species Latin names have been given, consulting the International Plant Names Index database (IPNI 2018). Numbered Voucher Specimens are available in the QAP Herbarium. There were 97 identified taxa, 27 of which are cultivated in any region of Ecuador after de la Torre et al. [29]. One collection of 97 duplicates and one collection of 97 numbered pictures of plants, without names, was prepared to be used in the workshops. They were labeled with numbers and symbols (+ if cultivated) as follows:

Aciotis purpurascens (Aubl.) Triana 58; Adenostemma fosbergii R.M. King & H. Rob. 94; Agonandra sp. 74; Alchornea triplinervia (Spreng.) Müll. Arg. 40; Annona muricata L. 4; Apeiba aspera Aubl. 68; Aspidosperma excelsum Benth. 83; Bauhinia tarapotensis Benth. 45; Bellucia pentamera Naudin 59; Besleria sp. 91; Calathea lutea (Aubl.) Schult. 35; Cecropia engleriana Snethl. 56; Cecropia ficifolia Warb. ex Snethl. 57; Chelonanthus alatus (Aubl.) Pulle8 4; Clarisia racemosa Ruiz & Pav. 53; Clidemia dentata Pav. ex D. Don 60; Clidemia octona (Bonpl.) L.O. Williams 61; Compsoneura sprucei (A. DC.) Warb. 6; Conyza sumatrensis (Retz.) E. Walker 95; Cordia alliodora (Ruiz & Pav.) Oken 93; Costus scaber Ruiz & Pav. 28; Croton lecheri Müll. Arg. 41; Cyathula prostrata (L.) Blume 75; Cyclanthus bipartitus Poit. ex A. Rich. 14; Danaea ulei Christ 2; Erechtites hieraciifolius (L.) Raf. ex DC.96; Erythrina poeppigiana (Walp.) O.F. Cook 46; Garcinia macrophylla Mart. 39; Graffenrieda gracilis (Triana) L.O. Williams 62; Guatteria multinervis Wall. 5; Heliconia chartacea Lane ex Barreiros 29; Heliconia episcopalis Vell. 30; Heliconia hirsuta L. f. 31; Heliconia rostrata Ruiz & Pav. 32; Heliconia shumanniana Loes. 33; Heliconia velutina L. Andersson 34; Homalomena crinipes Engl. 8; Homalomena picturata (Linden & André) Regel 9; Hyptis obtusiflora C. Presl ex Benth. 92; Inga alba (Sw.) Willd. 47; Inga auristellae Harms 48; Inga sapindoides Willd. 50; Jacaranda copaia (Aubl.) D. Don 90; Justicia comata (L.) Lam. 89; Leandra catequensis Gleason 63; Lunania parviflora Spruce ex Benth. 44; Miconia aureoides Cogn.64; Miconia paleacea Cogn. 65; Miconia punctata (Desr.) D. Don ex DC. 66; Minquartia guianensis Aubl. 73; Ochroma pyramidale (Cav. ex Lam.) Urb. 69; Paspalum pilosum Lam. 23; Perebea guianensis Aubl. 54; Perebea xanthochyma H. Karst. 55; Philodendron schmidtiae Croat & Cerón 10; Phytolacca sp. 76; Piptadenia sp. 52; Piptocoma discolor (Kunth) Pruski 97; Rhynchospora radicans (Schltdl. & Cham.) H. Pfeiff. 25; Scleria melaleuca Rchb. ex Schltdl. & Cham. 26; Selaginella exaltata (Kunze) Spring 1.; Siparuna sp. 3; Spermacoce exilis (L.O. Williams) C.D. Adams 85; Spermacoce remota Lam. 86; Tetracera volubilis L. 38; Tripogandra serrulata (Vahl) Handlos 27; Uncaria guianensis (Aubl.) J.F. Gmel. 87; Vismia baccifera (L.) Triana & Planch. 43; Witheringa solanacea L'Hér. 82; Xanthosoma saggitifolium (L.) Schott 11.

Ananas comosus (L.) Merr. +21; Ananas lucidus Mill. +22; Aphandra natalia (Balslev & A.J. Hend.) Barfod +15; Bactris gasipaes Kunth +16; Bixa orellana L. +67; Capsicum sp. +79; Carica papaya L. +72; Dioscorea trifida L.f. +12; Geonoma macrostachys Mart. +17; Gustavia longifolia Poepp. ex O. Berg +77; Inga edulis Mart. +49; Iriartea deltoidea Ruiz & Pav. +18; Lonchocarpus utilis A.C. Sm. +51; Manihot esculenta Crantz +42; Mauritia flexuosa L. f. +19; Musa acuminata Colla +36; Nicotiana tabacum L. +80; Oenocarpus batatua Mart. +20; Pouteria caimito (Ruiz & Pav.) Radlk. +78; Saccharum officinarum L.+ 24; Solanum quitoense Lam. + 81; Theobroma cacao L. + 70; Theobroma subincanum

Mart. +71; Warszewiczia coccinea (Vahl) Klotzsch +88; Zingiber officinale Roscoe 37; Carludovica palmata Ruiz & Pav. +13; Colocasia sculenta (L.) Schott +7.

## *2.3. Workshop Design*

Selection of the People and Participation

The workshop was carried out with a selected proportion of persons of Pakayaku who had knowledge and experience in plants. Kichwa language speakers were needed for translation. The selection of the people who participated was made after a dialog between one researcher (C.X.L.-Q r,g), members of the indigenous community, and the traditional authorities. The Kichwa president sent an invitation to each of the 20 people (n = 10 men; n = 10 women) chosen. To ensure a comfortable atmosphere, the workshop was held in the house of the President. The community was asked for logistical support for cleaning and preparing food in the workshop, and the participants received a bonus. The workshop was developed in a cordial and cheerful atmosphere, which essential in this type of research methodology so that the participants collaborate and that the data are reliable. Men and women were called at different dates so as not to coincide. The workshop took two days.

The activities consisted of showing specimens or photos of plants and asking the participants for their knowledge about them. One researcher (C.X.L.-Q.) wrote the answers on paper cards with the model design of Table 1. Twelve categories of uses have been previously described [26]. Votes and consensus on the categories of uses were communicated orally or by a show of hands. The researcher wrote down the information on the paper card. The filling process took 5–10 min, per plant, so 1 h was needed to make about 10 cards. Several sessions were performed to complete the total number separately (men and women). Finally, paper cards were transported by canoe and road to the Botany Laboratory at University for data digitalization.

**Table 1.** Paper card design, with data example filled in red.

| Reference Number: Plant 1 (See List in Materials and Methods) | | | |
|---|---|---|---|
| N° People that Consider it Useful Plant: 10 Sex: Men | | | |
| Category | N° Persons Who Cite it in: | Select the Most Frequent Use (Only 1) *& Sign as +F* | Select the Preferred Use (Only 1) *& Sign it as +P* |
| 1 Food for human | 10 | +F | |
| 2 Food for animal | | | |
| 3 Utensils and tools | | | |
| 4 Handcrafts | | | |
| 5 Construction | 3 | | |
| 6 Cultural uses | | | |
| 7 Human medicine | 10 | | +P |
| 8 Veterinary uses | 9 | | |
| 9 Poisonous plants | | | |
| 10 Ornamental plants | | | |
| 11 Environmental uses | | | |
| 12 Plants for fuel | | | |

## *2.4. Data Digitalization*

Row data from the paper cards were introduced in an Excel Spreadsheet (Microsoft 2013) like the one shown in Figure 3. The rows contain the 97 scientific names and two repeated blocks of columns with numbers 1–12, corresponding to the 12 categories of uses considered. The cells of the left block contain the data obtained from the men, and the cells of the right block the data are from the women.

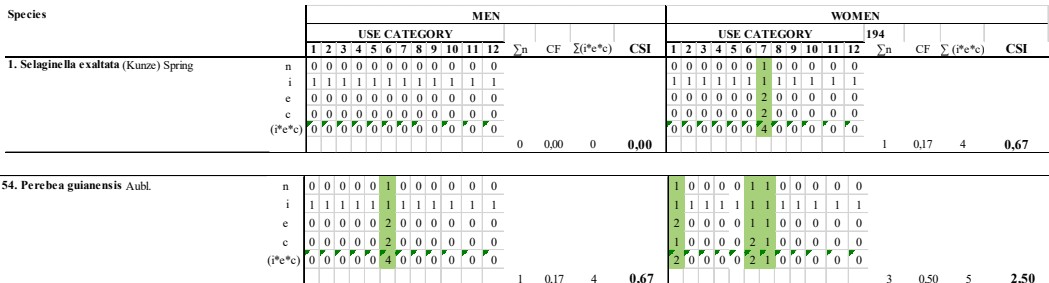

**Figure 3.** Excel spreadsheet model to introduce row data from paper cards.

The cells were filled in the following way:

- Row n: Proportion of the workshop participants who handed up for that use category. It is expressed as per participant, not in percentage. Minimum value 0. Maximum value 1 (see Figure 4 for transcription of the example of Table 1).
- Row i (management): Significance of the plant from an agronomical point of view. For wild plants, value, = 1. For plants referenced as cultivated in the *Encyclopedia of Useful Plants of Ecuador* [29] value, = 2.
- Row e (preference): Proportion of the workshop participants who selected that use category as "the most preferred". It is expressed as per participant, not in percentage. Maximum value 2 was given to the preferred category. Non-preferred categories value = 1. Non-used categories = 0.
- Row c (frequency): Proportion of the workshop who selected that use category as "the most frequently used in the community". It is expressed as per participant, not in percentage. Maximum value 2 was given to the selected category. Non-selected categories value = 1. Non-used categories = 0.
- Row (i*c*e): Value obtained for a category of use of the species in the workshop. Maximum value $(2 \times 2 \times 2) = 8$. Minimum value $(1 \times 1 \times 1) = 1$ Non-used categories = 0.
- Column $\sum$n: Minimum value 0. maximum value = 12. H = the highest value obtained in the compared workshops by one species.
- Column $\sum$(i*e*c): Global value of all the uses of the species. The maximum value that one species can obtain is $((2 \times 2 \times 2) + [11 \times (2 \times 1 \times 1)]) = 30$.

| | | MEN | | | | | | | | | | | | |
|---|---|---|---|---|---|---|---|---|---|---|---|---|---|---|
| | | Use category | | | | | | | | | | | | |
| | | 1 | 2 | 3 | 4 | 5 | 6 | 7 | 8 | 9 | 10 | 11 | 12 | $\sum$n |
| Plant 1 | n | 1.0 | 0 | 0 | 0 | 0.3 | 0 | 1.0 | 0.9 | 0 | 0 | 0 | 0 | 3.2 |

**Figure 4.** Example data from Table 1 (paper card) transcribed to a spreadsheet structure (first row of Figure 3).

### 2.5. Quantitative Analysis of Data: Cultural Significance Indexes (CSI) Calculations and Statistics

After a critical review of the applicable indices had been carried out [30], we selected the CSI of Silva et al., 2006 [31] but propose some modifications from its original mathematical function

The CSI [31] assesses the relevance that a species can have for an informant among a set of species. In its original design, it is not necessary to have a pre-established classification of uses. The informants can classify them as they decide during the interview.

Our proposal, apart from considering a closed list of 12 types of uses of Luzuriaga [26], calculates the value of the H parameter, following its previously mentioned definition. The mathematical function to calculate modified CSI is as follows:

$$\text{CSI} = \text{CF} \times \left( \sum (i*e*c)_i \right) = \left( \sum n/H \right) \times \left( \sum (i*e*c)_i \right)$$

where n = use; I = management; e = preference; c = frequency; and correcting factor (CF) = $\sum$n/H.

In theory, any species might reach the maximum variety of uses, which was 12 in our classification system. However, it is well known that, in practice, the maximum number of uses of one species is lower. It depends on the plant and on the knowledge of the participants of a concrete workshop. In order to compare the results obtained in a different workshop, we employed the CF as defined above. This CF modulates the knowledge about one species uses in the context of the proper workshop. It is very useful when we need to measure or compare between species or group of species and informants or groups of informants. CF maximum value = 1. CF measures the degree of deviation of the species uses to the most useful situation found in that workshop, so it allows interpretation of the results of the questionnaire in the context of the workshop itself. For this reason, it is a very important parameter to distinguish the knowledge of two replicated groups: the group of women and the group of men. It is an intra-quantitative comparative parameter.

CSI is the total sum of different types of uses of the plant expressed by numbers. It helps to order lists of species upon the importance of their utility and to compare them. It considers not just the frequency of use but also a subjective valorization of quality, indirectly measured through the parameter e (= preferred category). Management places much weight on this formula because it is considered that knowing or ignoring a wild plant has a very different cultural than agroforestry explanation. This appreciation is influenced by the former definition of CSI from Silva et al. [31].

Individual CSI values of the 97 plants under the perspective of the 12 use categories were obtained. Knowledge from men was quantified as the CSI men summary and abbreviated as CSIm. The women's knowledge, abbreviated as CSIw, was calculated the same way.

Non-parametric statistical tests were applied to compare groups of CSI values (Wilcoxon test, with SPSS V.20 for Windows).

## 3. Results

The results are summarized in Table 2. Men recognized 135 uses of 77 plants, and women recognized 294 uses of 92 plants. The highest number of use categories assigned to one plant (H) was six. When comparing by Wilcoxon text, the CSI of men for n = 97 taxa, with the CSI of women for the same taxa, statistically significant differences were found (n = 97, *** $p < 0.001$). The total value of CSIm and CSI were 164.67 and 267.50, respectively. Row data can be consulted in Table S1 of the Supplementary Materials.

**Table 2.** The 97 studied taxa, ordered by the APG IV classification system. $\sum$n, number of uses; cultural significance index (CSI). Men (CSIm) in black, and women (CSIw) in violet. The highest CSI values for each species are in grey cells.

| | Men | | Women | |
|---|---|---|---|---|
| | $\sum$n | CSIm | $\sum$n | CSIw |
| 1. *Selaginella exaltata* **(Kunze) Spring** | 0 | 0.00 | 1 | 0.67 |
| 2. *Danaea ulei* **Christ** | 1 | 0.67 | 0 | 0.00 |
| 3. *Siparuna* **sp.** | 2 | 1.33 | 3 | 3.00 |
| 4. *Annona muricata* **L.** | 1 | 1.33 | 2 | 3.33 |
| 5. *Guatteria multinervis* **Wall.** | 3 | 3.00 | 3 | 3.00 |
| 6. *Compsoneura sprucei* **(A. DC.) Warb.** | 1 | 2.00 | 1 | 0.67 |
| 7. *Colocasia sculenta* **(L.) Schott** | 3 | 6.00 | 3 | 6.00 |
| 8. *Homalomena crinipes* **Engl.** | 2 | 1.67 | 3 | 3.00 |
| 9. *Homalomena picturata* **(Linden & André) Regel** | 2 | 1.67 | 3 | 3.00 |
| 10. *Philodendron schmidtiae* **Croat & Cerón** | 1 | 0.67 | 2 | 1.67 |
| 11. *Xanthosoma saggitifolium* **(L.) Schott** | 2 | 1.67 | 2 | 1.67 |
| 12. *Dioscorea trifida* **L.f.** | 2 | 3.33 | 2 | 3.33 |
| 13. *Carludovica palmata* **Ruiz & Pav.** | 2 | 3.33 | 6 | 18.00 |
| 14. *Cyclanthus bipartitus* **Poit. ex A. Rich.** | 2 | 1.33 | 4 | 4.00 |

**Table 2.** *Cont*.

| | Men | | Women | |
|---|---|---|---|---|
| | ∑n | CSIm | ∑n | CSIw |
| 15. *Aphandra natalia* (Balslev & A.J. Hend.) Barfod | 2 | 2.67 | 4 | 8.00 |
| 16. *Bactris gasipaes* Kunth | 4 | 8.00 | 3 | 6.00 |
| 17. *Geonoma macrostachys* Mart. | 1 | 1.33 | 3 | 6.00 |
| 18. *Iriartea deltoidea* Ruiz & Pav. | 2 | 3.33 | 6 | 18.00 |
| 19. *Mauritia flexuosa* L. f. | 2 | 3.33 | 3 | 6.00 |
| 20. *Oenocarpus batatua* Mart. | 4 | 9.33 | 6 | 18.00 |
| 21. *Ananas comosus* (L.) Merr. | 0 | 0.00 | 2 | 3.33 |
| 22. *Ananas lucidus* Mill. | 3 | 6.00 | 2 | 1.33 |
| 23. *Paspalum pilosum* Lam. | 0 | 0.00 | 0 | 0.00 |
| 24. *Saccharum officinarum* L. | 2 | 3.33 | 2 | 3.33 |
| 25. *Rhynchospora radicans* (Schltdl. & Cham.) H. Pfeiff. | 0 | 0.00 | 0 | 0.00 |
| 26. *Scleria melaleuca* Rchb. ex Schltdl. & Cham. | 0 | 0.00 | 1 | 0.67 |
| 27. *Tripogandra serrulata* (Vahl) Handlos | 1 | 0.67 | 1 | 0.67 |
| 28. *Costus scaber* Ruiz & Pav. | 2 | 1.67 | 3 | 2.50 |
| 29. *Heliconia chartacea* Lane ex Barreiros | 1 | 0.67 | 1 | 0.67 |
| 30. *Heliconia episcopalis* Vell. | 2 | 1.67 | 2 | 1.33 |
| 31. *Heliconia hirsuta* L. f. | 0 | 0.00 | 1 | 0.67 |
| 32. *Heliconia rostrata* Ruiz & Pav. | 2 | 1.67 | 2 | 1.33 |
| 33. *Heliconia shumanniana* Loes. | 0 | 0.00 | 2 | 1.67 |
| 34. *Heliconia velutina* L. Andersson | 1 | 0.67 | 1 | 0.67 |
| 35. *Calathea lutea* (Aubl.) Schult. | 1 | 0.67 | 1 | 0.67 |
| 36. *Musa acuminata* Colla | 1 | 1.33 | 3 | 6.00 |
| 37. *Zingiber officinale* Roscoe | 1 | 1.33 | 2 | 3.33 |
| 38. *Tetracera volubilis* L. | 0 | 0.00 | 1 | 0.67 |
| 39. *Garcinia macrophylla* Mart. | 2 | 1.67 | 3 | 3.00 |
| 40. *Alchornea triplinervia* (Spreng.) Müll. Arg. | 1 | 0.67 | 1 | 0.67 |
| 41. *Croton lecheri* Müll. Arg. | 2 | 1.67 | 2 | 1.67 |
| 42. *Manihot esculenta* Crantz | 2 | 3.33 | 2 | 3.33 |
| 43. *Vismia baccifera* (L.) Triana & Planch. | 0 | 2.00 | 2 | 1.67 |
| 44. *Lunania parviflora* Spruce ex Benth. | 1 | 0.67 | 4 | 4.00 |
| 45. *Bauhinia tarapotensis* Benth. | 4 | 4.67 | 2 | 1.67 |
| 46. *Erythrina poeppigiana* (Walp.) O.F. Cook | 1 | 0.67 | 3 | 3.00 |
| 47. *Inga alba* (Sw.) Willd. | 3 | 2.50 | 2 | 1.33 |
| 48. *Inga auristellae* Harms | 3 | 3.00 | 2 | 1.33 |
| 49. *Inga edulis* Mart. | 2 | 3.33 | 3 | 6.00 |
| 50. *Inga sapindoides* Willd. | 3 | 2.50 | 2 | 1.67 |
| 51. *Lonchocarpus utilis* A.C. Sm. | 1 | 1.33 | 0 | 0.00 |
| 52. *Piptadenia* sp. | 1 | 0.67 | 2 | 1.67 |
| 53. *Clarisia racemosa* Ruiz & Pav. | 2 | 1.67 | 2 | 1.67 |
| 54. *Perebea guianensis* Aubl. | 1 | 0.67 | 3 | 2.50 |
| 55. *Perebea xanthochyma* H. Karst. | 2 | 1.67 | 2 | 1.67 |
| 56. *Cecropia engleriana* Snethl. | 0 | 0.00 | 1 | 0.67 |
| 57. *Cecropia ficifolia* Warb. ex Snethl. | 1 | 0.67 | 2 | 2.00 |
| 58. *Aciotis purpurascens* (Aubl.) Triana | 0 | 0.00 | 1 | 0.67 |
| 59. *Bellucia pentamera* Naudin | 1 | 0.67 | 1 | 0.67 |
| 60. *Clidemia dentata* Pav. ex D. Don | 1 | 0.67 | 1 | 0.67 |
| 61. *Clidemia octona* (Bonpl.) L.O. Williams | 1 | 0.00 | 1 | 0.67 |
| 62. *Graffenrieda gracilis* (Triana) L.O. Williams | 1 | 0.00 | 1 | 0.67 |
| 63. *Leandra catequensis* Gleason | 0 | 0.00 | 1 | 0.67 |
| 64. *Miconia aureoides* Cogn. | 1 | 0.67 | 2 | 1.33 |
| 65. *Miconia paleacea* Cogn. | 1 | 0.67 | 1 | 0.67 |
| 66. *Miconia punctata* (Desr.) D. Don ex DC. | 1 | 0.67 | 3 | 2.50 |
| 67. *Bixa orellana* L. | 4 | 9.33 | 4 | 9.33 |
| 68. *Apeiba aspera* Aubl. | 1 | 0.67 | 1 | 0.67 |

**Table 2.** *Cont.*

| | Men | | Women | |
|---|---|---|---|---|
| | $\sum$n | CSIm | $\sum$n | CSIw |
| 69. *Ochroma pyramidale* (Cav. ex Lam.) Urb. | 2 | 1.67 | 5 | 7.50 |
| 70. *Theobroma cacao* L. | 2 | 2.67 | 2 | 3.33 |
| 71. *Theobroma subincanum* Mart. | 3 | 6.00 | 3 | 6.00 |
| 72. *Carica papaya* L. | 2 | 3.33 | 3 | 5.00 |
| 73. *Minquartia guianensis* Aubl. | 1 | 0.67 | 2 | 1.33 |
| 74. *Agonandra* sp. | 1 | 0.67 | 3 | 3.00 |
| 75. *Cyathula prostrata* (L.) Blume | 1 | 0.67 | 0 | 0.00 |
| 76. *Phytolacca* sp. | 0 | 0.00 | 2 | 1.33 |
| 77. *Gustavia longifolia* Poepp. ex O. Berg | 1 | 1.33 | 1 | 1.33 |
| 78. *Pouteria caimito* (Ruiz & Pav.) Radlk. | 3 | 6.00 | 2 | 3.33 |
| 79. *Capsicum* sp. | 3 | 6.00 | 4 | 9.33 |
| 80. *Nicotiana tabacum* L. | 1 | 1.33 | 1 | 1.33 |
| 81. *Solanum quitoense* Lam. | 2 | 3.33 | 3 | 6.00 |
| 82. *Witheringa solanacea* L'Hér. | 0 | 0.00 | 1 | 0.67 |
| 83. *Aspidosperma excelsum* Benth. | 1 | 0.67 | 1 | 0.67 |
| 84. *Chelonanthus alatus* (Aubl.) Pulle | 0 | 0.00 | 1 | 0.67 |
| 85. *Spermacoce exilis* (L.O. Williams) C.D. Adams | 0 | 0.00 | 1 | 0.67 |
| 86. *Spermacoce remota* Lam. | 1 | 0.67 | 1 | 0.67 |
| 87. *Uncaria guianensis* (Aubl.) J.F. Gmel. | 1 | 0.67 | 3 | 4.00 |
| 88. *Warszewiczia coccinea* (Vahl) Klotzsch | 1 | 1.33 | 2 | 2.67 |
| 89. *Justicia comata* (L.) Lam. | 0 | 0.00 | 1 | 0.67 |
| 90. *Jacaranda copaia* (Aubl.) D. Don | 2 | 1.33 | 2 | 1.33 |
| 91. *Besleria* sp. | 0 | 0.00 | 1 | 0.67 |
| 92. *Hyptis obtusiflora* C. Presl ex Benth. | 0 | 0.00 | 1 | 0.67 |
| 93. *Cordia alliodora* (Ruiz & Pav.) Oken | 1 | 0.67 | 1 | 0.67 |
| 94. *Adenostemma fosbergii* R.M. King & H. Rob. | 1 | 0.67 | 1 | 0.67 |
| 95. *Conyza sumatrensis* (Retz.) E. Walker | 0 | 0.00 | 1 | 0.67 |
| 96. *Erechtites hieraciifolius* (L.) Raf. ex DC. | 1 | 0.67 | 1 | 0.67 |
| 97. *Piptocoma discolor* (Kunth) Pruski | 4 | 4.00 | 3 | 2.50 |
| **TOTAL** | 135 | 164.67 | 294 | 267.50 |

The 25 most important plants for the studied community are shown in Table 3.

**Table 3.** The 25 studied taxa ordered by decreasing values of CSIm + CSIw. Those species with CSIw ≥ CSIm appear in grey filled cells. CSIm = CSI for men (in black); CSIw = CSI for women (in violet) + = cultivated taxa in Ecuador after [29].

| | CSIm | CSIw | CSIm + CSIw |
|---|---|---|---|
| +20. *Oenocarpus batatua* Mart. | 9.33 | 18.00 | 27.33 |
| +13. *Carludovica palmata* Ruiz & Pav. | 3.33 | 18.00 | 21.33 |
| +18. *Iriartea deltoidea* Ruiz & Pav. | 3.33 | 18.00 | 21.33 |
| +67. *Bixa orellana* L. | 9.33 | 9.33 | 18.67 |
| +79. *Capsicum* sp. | 6.00 | 9.33 | 15.33 |
| +16. *Bactris gasipaes* Kunth | 8.00 | 6.00 | 14.00 |
| +7. *Colocasia sculenta* (L.) Schott | 6.00 | 6.00 | 12.00 |
| +71. *Theobroma subincanum* Mart. | 6.00 | 6.00 | 12.00 |
| +15. *Aphandra natalia* (Balslev & A.J. Hend.) Barfod | 2.67 | 8.00 | 10.67 |
| +78. *Pouteria caimito* (Ruiz & Pav.) Radlk. | 6.00 | 3.33 | 9.33 |
| +19. *Mauritia flexuosa* L. f. | 3.33 | 6.00 | 9.33 |
| +49. *Inga edulis* Mart. | 3.33 | 6.00 | 9.33 |
| +81. *Solanum quitoense* Lam. | 3.33 | 6.00 | 9.33 |
| +69. *Ochroma pyramidale* (Cav. ex Lam.) Urb. | 1.67 | 7.50 | 9.17 |

**Table 3.** *Cont.*

|  | CSIm | CSIw | CSIm + CSIw |
|---|---|---|---|
| +72. *Carica papaya* L. | 3.33 | **5.00** | 8.33 |
| 22. *Ananas lucidus* Mill. | 6.00 | **1.33** | 7.33 |
| +17. *Geonoma macrostachys* Mart. | 1.33 | **6.00** | 7.33 |
| +36. *Musa acuminata* Colla | 1.33 | **6.00** | 7.33 |
| 12. *Dioscorea trifida* L.f. | 3.33 | **3.33** | 6.67 |
| +24. *Saccharum officinarum* L. | 3.33 | **3.33** | 6.67 |
| +42. *Manihot esculenta* Crantz | 3.33 | **3.33** | 6.67 |
| 97. *Piptocoma discolor* (Kunth) Pruski | 4.00 | **2.50** | 6.50 |
| 45. *Bauhinia tarapotensis* Benth. | 4.67 | **1.67** | 6.33 |
| 5. *Guatteria multinervis* Wall | 3.00 | **3.00** | 6.00 |
| +70. *Theobroma cacao* L. | 2.67 | **3.33** | 6.00 |

## 4. Discussion

Some authors [24,27,32,33] have described, from an anthropological point of view, notable differences in the social and daily lives of males and females of the Kichwa communities. The male role is associated with hunting and fishing. Women are dedicated to the cultivation of the farms, personal beautification, the attention and care of the children, the preparation of food and drinks, and the elaboration of ceramics. This division of the tasks leads to the initial hypothesis regarding the knowledge of plants of males and females.

The analysis of the results of our workshops reveals that women have a greater knowledge of the diversity and uses of the categories and subcategories of plant uses that are related to the divisions of work in society. Therefore, they are depositaries of a good part of the knowledge of medicinal plants, which is also related to their role in the care and attention of children and family in daily life. Plants are used by the community [26] for blows, cuts, bites, inflammations, skin and bones, and pains in general. The most important masculine activities are the capture of monkeys, birds, armadillos, deer, felines, and bears and fishing for aquatic snails, lizards, turtles, and frogs [32]. They are still using spears, bodoqueras, and darts impregnated with curare, although today hunting is done with shotguns. In any case, this practice is being reduced [24]. Men also knock down trees for building houses and canoes and for firewood. They are well-related to our experimental results. Finally, we must point that our data suggest that the decline of traditional knowledge related to plant diversity associated with the male population can be related to the fact that traditional knowledge of the men was formerly about hunting and fishing, and hunting is much less practiced than before.

The quantitative analysis carried out has demonstrated the relevance of the role of the Canelo-Kichwa woman as a leading agent of the conservation of the traditional knowledge of the plants in the community. We have found statistically significant differences ($p = 0.000$, n = 97, Wilcoxon CSI test), which should serve to value the function they develop in society. An important aspect to take into account is the valorization of their activities among the population; otherwise, additional knowledge will be lost. This can lead to a decline of original cultures and this is especially important in large areas located to the south of the province of Pastaza [34]. In these areas, a clear westernization has produced a strong change of customs in the local communities [35,36]. In recent decades, this has been accelerated by the construction of highways, roads, and local airports by oil companies.

The United Nations SGD2, is interested in maintaining the genetic diversity of seeds, cultivated plants, farmed and domesticated animals, and their related wild species. Research projects in the environment, agriculture, and agroindustry thematic that allow the agricultural production of our studied area are being requested by the scientific community [7,36]. In the specific area of medicinal plants, some authors [37] have already performed a meta-analysis of the effect of gender on knowledge. Thus, to meet SGD2 in tropical agriculture systems, we must reinforce the importance of considering the gender perspective.

## 5. Conclusions

The causes of the difference knowledge and cultural significance that we have demonstrated quantitatively are in accordance with population characteristics, the structure of society, and the Kichwa worldview. The consequence of these assessments is that the gender perspective must be taken into account when carrying out ethnobotanical studies in these communities. In addition, they must be considered in a very special way in biodiversity conservation programs and the rescue of traditional knowledge related to it. To ensure sustainable food production systems and to implement resilient agricultural practices that increase productivity and strengthen the capacity for adaptation to change, all of these premises must be taken into account.

**Supplementary Materials:** The following are available online at http://www.mdpi.com/2071-1050/11/15/4211/s1, Table S1: Row data calculations for CSIm and CSIw (Cultural Significance Indexes for men and women, respectively) of the 97 plants used in our workshops

**Author Contributions:** Conceptualization, T.R.-T.; Methodology, C.E.C.-M. and C.X.L.-Q.; Validation J.B.-S.; Formal Analysis, M.H.d.B.; Investigation, C.X.L.-Q.; Data Curation, C.E.C.-M. and C.X.L.Q.; Writing—Original Draft Preparation, T.R.-T.; Writing—Review and Editing, J.B.-S.; Visualization, M.H.d.B.; Supervision, C.X.L.-Q.; Project Administration, T.R.-T.; Funding Acquisition, C.X.L.-Q. and T.R.-T.

**Funding:** This research was partially funded by the Government of Extremadura (Spain) and the European Union through the action "Apoyos a los Planes de Actuación de los Grupos de Investigación Catalogados de la Junta de Extremadura: FEDER GR15080".

**Acknowledgments:** The authors would like to extend their sincere thanks to the members of the Kichwa community of Pakayaku, Luzmila Gayas, the People's Assembly of Pakayaku and the collaborating ayllus (families), for their cooperation during the fieldwork.

**Conflicts of Interest:** The authors declare no conflict of interest.

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
