# Peer review of "Plant Biodiversity Knowledge Varies by Gender in Sustainable Amazonian Agricultural Systems Called Chacras"

_sustainability, doi:10.3390/su11154211_

Round 1

Reviewer 1 Report

 I want to make some appreciations about this paper:

In the Abstract, on line 27 indicate that it refers to "The United Nations Sustainable Development Goal 2 (SGD2)"

I do not see clearly if  "Parcela Borde del río, Parcela Igapo and Parcela colinas" (yellow boxes of Figure 1) are included in this study. I think you should eliminate them.

Place the columns correctly in the paper card design (figure 2).

In figure 4, it is not clear where the values of the table come from.

There is a typographical error on line 294: bites are repeated

Author Response

I want to make some appreciations about this paper:

1.      In the Abstract, on line 27 indicate that it refers to "The United Nations Sustainable Development Goal 2 (SGD2)" OK. We have included this.

2.      I do not see clearly if  "Parcela Borde del río, Parcela Igapo and Parcela colinas" (yellow boxes of Figure 1) are included in this study. I think you should eliminate them. Ok. We have changed the Figure.

3.      Place the columns correctly in the paper card design (figure 2). Ok. We have  changed the figure

4.      In figure 4, it is not clear where the values of the table come from. OK The caption of figure 4 has been changed. 

5.      There is a typographical error on line 294: bites are repeated. OK.  It has been deleted.

Reviewer 2 Report

In this study, the authors assessed traditional plant knowledge in an isolated community in Amazonian Ecuador. They focus on the difference between males and females in their knowledge on plant diversity and plant uses. This is a beautiful work, with a well-considered and well-prepared methodology. Although I am enthusiastic about the work, I have some concerns that I think should be addressed before the manuscript could be considered for publication.

The introduction starts with the global food- and biodiversity crisis, and then zooms-in on a rotating agricultural system (the Chacras) in a remote area in Ecuador. This structure does not work for me. Such systems may be sustainable in remote areas with low human population densities, but are not a solution for food security on larger scales. In addition, biodiversity is high in such systems because they are relatively small and surrounded by large areas of forest. Thus these fields are continuously invaded by non-agricultural species, and probably half forest / half field for most of the time. On larger scales, that balance, and thus biodiversity, will change. I strongly suggest placing the study in another context, for example to focus only on the importance to consider a gender perspective in ethnobotanical studies.

The manuscript needs to be revised by a native speaker. Grammatical errors are very numerous and in many places unfortunately hinder the interpretation of the text (I have highlighted some errors below, but they are too numerous to point out). In addition, I found the introduction and discussion quite unclear. For many paragraphs, I was left wondering what exactly the authors are trying to point out, or conclude (see some detailed comments below).

L20. “potential reservoirs of the traditional biodiversity of chacras plant knowledge”. The word “biodiversity” is not correct here. Suggest using “potential reservoirs of traditional chacras plant knowledge”.

L22. “Uncontacted” is misleading, and not true. Suggest replacing with “isolated” here, and throughout the manuscript.

L41. “forests are bringing down while greenhouse-gas emissions grow up on a large scale” should be something like “deforestation is increasing while greenhouse-gas emissions continuous to increase”.

L47. “up to 40%” in the Amazon?

L52. “identity” should be “identities”.

L59. “interaction between the indigenous woman lifehood” and what other factor?

L61. Suggest using other word for “diagnostic”.

L66. Should “food plains” be “floodplains”?

L81. Suggest using other words for “dominate the forests”.

L96. “communication” should be “connection”.

L97. “originary” should be “original”.

L102. Suggest explaining what is meant with “Pakayaku access is not free.”.

L128. “50x2” add unit (meters?); “method” can be removed here.

L132-133. Suggest removing; this is not relevant information.

L137. What do the authors mean with “cultivated in any region of Ecuador upon de la Torre et al.[29].”?

L87-88. “One attendee..” and L89-90 “One researcher..” Is the same said twice here? If so, I suggest removing the first sentence. If not, what is meant in L87-88?

L198, 201, 209. Should “raw” be “row”?

L206, 213, 216. What do the authors mean with “It is expressed as a both, not in percentage.”?

L261-263. Suggest adding explanation of why management has so much more weight.

L273. “p=0.000” should be “p<0.001”.

L289. Suggest replacing “of the male and female collective” with “of males and females”.

L290-291. “of the categories and subcategories that are related to the divisions of work in society”. This expression is very vague. Do the authors not just mean “of plants”.

L292-293. Suggest replacing “plants, related to the care” with “plants, which is also related to their role in the care”. And add “are” before “used” in the next sentence.

L294. “bites” is mentioned twice here.

L295. What do the authors mean with “by antonomasia”?

L299-302. “Finally we must point that our data suggest that the decline of traditional knowledge related to plant diversity associated with the male population can be related with the fact that hunting is nowadays less practiced than before.” I do not understand what the authors are trying to point out here and how the data would show traditional knowledge is lost.

L303. “quantitative” is mention twice here.

L307-308. “An important aspect to take into account is the valorization of their activities among the population. On the contrary, that traditional knowledge will be lost.” I again do not fully understand what the authors try to point out here, or where the “on the contrary” refers to.

Author Response

In this study, the authors assessed traditional plant knowledge in an isolated community in Amazonian Ecuador. They focus on the difference between males and females in their knowledge on plant diversity and plant uses. This is a beautiful work, with a well-considered and well-prepared methodology. Although I am enthusiastic about the work, I have some concerns that I think should be addressed before the manuscript could be considered for publication. Thank you very much for your appreciations.

The introduction starts with the global food- and biodiversity crisis, and then zooms-in on a rotating agricultural system (the Chacras) in a remote area in Ecuador. This structure does not work for me. We understand what you mean. You explain perfectly in your position in the following paragraphs.: Such systems may be sustainable in remote areas with low human population densities, but are not a solution for food security on larger scales. In our perspective, global food security has to take in account not only these large scales that you mention but also the scale we are dealing with.  Because, it is also large. Very large. Amazonian populations must be considered because of its social,  ethnobiological, and environmental importance. It is a mistake to forget in our analysis Amazonian reality. 

 In addition, biodiversity is high in such systems because they are relatively small and surrounded by large areas of forest. Thus these fields are continuously invaded by non-agricultural species, and probably half forest / half field for most of the time. On larger scales, that balance, and thus biodiversity, will change.  You are right but we have focused our effort not to the numbers or quantity but the cualitative importance. Our idea is reinforced by the urgencies of the SGD-

I strongly suggest placing the study in another context, for example to focus only on the importance to consider a gender perspective in ethnobotanical studies. We understand what you suggest but we would like to emphasize on the importance of connecting this management practices with food soveringty. This is very important for the local populations. Even though, we have deleted some parts of the text in order to attend your claim as much as possible.

The manuscript needs to be revised by a native speaker. Grammatical errors are very numerous and in many places unfortunately hinder the interpretation of the text (I have highlighted some errors below, but they are too numerous to point out). In addition, I found the introduction and discussion quite unclear. For many paragraphs, I was left wondering what exactly the authors are trying to point out, or conclude (see some detailed comments below). Ok. We welcome this criticism in its entirety and to resolve it we have sent the manuscript to the Language Correcting Service of MPDI.  

·        L20. “potential reservoirs of the traditional biodiversity of chacras plant knowledge”. The word “biodiversity” is not correct here. Suggest using “potential reservoirs of traditional chacras plant knowledge”. Ok Changed,

·        L22. “Uncontacted” is misleading, and not true. Suggest replacing with “isolated” here, and throughout the manuscript. Ok Changed,

·        L41. “forests are bringing down while greenhouse-gas emissions grow up on a large scale” should be something like “deforestation is increasing while greenhouse-gas emissions continuous to increase”. Ok Changed,

·        L47. “up to 40%” in the Amazon?. Yes.

·        L52. “identity” should be “identities”.Ok Changed

·        L59. “interaction between the indigenous woman lifehood” and what other factor? Mistake. Changed.

·        L61. Suggest using other word for “diagnostic”. Ok. Description has been used.

·        L66. Should “food plains” be “floodplains”? Ok. Yes.

·        L81. Suggest using other words for “dominate the forests” Ok, Changed by “be specialized in”

·        L96. “communication” should be “connection”.Ok. Changed.

·        L97. “originary” should be “original”. Ok. Changed.

·        L102. Suggest explaining what is meant with “Pakayaku access is not free.”It means that you cannot get into if you are not allowed by the kuraka or the President of Pakayaku.  If you do it, you can be reduced or attacked.

·        L128. “50x2” add unit (meters?); “method” can be removed here. OK .

·        L132-133. Suggest removing; this is not relevant information. We prefer to maintain it because it is usually asked by Botanical Referees.

·        L137. What do the authors mean with “cultivated in any region of Ecuador upon de la Torre et al.[29].”? Cultivated in Ecuador, as it is said in the Enciclopedia de las Plantas Utiles de Ecuador, a book that has been written by those authors.

·        L87-88. “One attendee..” and L89-90 “One researcher..” Is the same said twice here? If so, I suggest removing the first sentence. If not, what is meant in L87-88? Ok. Changed.

·        L198, 201, 209. Should “raw” be “row”? Ok. Changed.

·        L206, 213, 216. What do the authors mean with “It is expressed as a both, not in percentage.”? Ok, -changed. Both per one

·        L261-263. Suggest adding explanation of why management has so much more weight.Ok. included.

·        L273. “p=0.000” should be “p<0.001”. Yes. Corrected.

·        L289. Suggest replacing “of the male and female collective” with “of males and females”. Ok. Changed.

·        L290-291. “of the categories and subcategories that are related to the divisions of work in society”. This expression is very vague. Do the authors not just mean “of plants”. Corrected. In means “of plant uses”.

·        L292-293. Suggest replacing “plants, related to the care” with “plants, which is also related to their role in the care”. And add “are” before “used” in the next sentence. Corrected.

·        L294. “bites” is mentioned twice here. Corrected.

·        L295. What do the authors mean with “by antonomasia”? Corrected, changed.

·        L299-302. “Finally we must point that our data suggest that the decline of traditional knowledge related to plant diversity associated with the male population can be related with the fact that hunting is nowadays less practiced than before.” I do not understand what the authors are trying to point out here and how the data would show traditional knowledge is lost. Better explanation included.

·        L303. “quantitative” is mention twice here. Corrected.

L307-308. “An important aspect to take into account is the valorization of their activities among the population. On the contrary, that traditional knowledge will be lost.” I again do not fully understand what the authors try to point out here, or where the “on the contrary” refers to. Ok, -sentence modified.

Reviewer 3 Report

1. The Enlgish should be improved, e.g. amazonic, kitchwa, latin – with capital

2. lie, rub and burn sound strange is it not better to write: slash-and-burn?

3. The paper lacks information on what are the specific uses of the studied plants.

4. There is a large literature on gender differences in knowledge od plants, including South America. It is not discussed here.

5. Voucher numbers in the herbarium are missing.

6. A map would be useful.

7. Some diagram showing the chakras system could help readers to imagine it.

8. There is no need to directly show every step of data input in Excel. It is usually not done in papers.

Author Response

1. The Enlgish should be improved, e.g. amazonic, kitchwa, latin – with capital. Ok Done The manuscript has been sent to MPDI Language Service..

2. lie, rub and burn – sound strange – is it not better to write: slash-and-burn? Ok. Changed.

3. The paper lacks information on what are the specific uses of the studied plants. Sorry but this is not the objective of the paper. We have included just the uses mentioned by the informants.

4. There is a large literature on gender differences in knowledge of plants, including South America. It is not discussed here. We have tried to improve bibliography on this topic and we have included it in this new version of the paper.

5. Voucher numbers in the herbarium are missing. We can include it in an Annex. They appear in the Ph Tesis of one of the autors C.X. Luzuriaga Quichimbo (2017). This bibliographic cite is included in the reference list. We did not include each voucher number in order to facilitate the reading. But the addition in an Annex is available upon request.

6. A map would be useful. OK. It has been included.

7. Some diagram showing the chakras system could help readers to imagine it. Ok It has been included.

8. There is no need to directly show every step of data input in Excel. It is usually not done in papers. We think it is necessary because the process is somehow tedious and it requires to be explained in detail as to favour the paper can be replicated anywhere.

Round 2

Reviewer 2 Report

The authors have properly addressed most previous comments. Although they give valid arguments for their choices, none of the main concerns raised previously have been addressed in the revision. I recommend that some of the arguments given in the reply to justify this are included in the introduction/discussion. In addition, in some cases (e.g. “Pakayaku access is not free.”), the unclarity remains in the text, and the authors only explain what they mean in the reply. This is of course not useful for other readers (who may also wonder what is meant with “free” in this context). I recommend to edit the text where unclarities were previously indicated, if not yet done so. The grammar also still needs to be revised. Lastly, I have a few minor suggestions left: (i) remove “perfect” in the first line of the abstract (this relates again to my previous comment on small-scale versus large-scale; on large scales such systems are unlikely sustainable), (ii) “primary forest” should be “secondary forest in figure 1, (iii) Figure 6 and 7 (and maybe all figures except figure 1 and 2) are in fact tables, and should thus be named "Table 1”, etc.

Author Response

The authors have properly addressed most previous comments. Although they give valid arguments for their choices, none of the main concerns raised previously have been addressed in the revision. I recommend that some of the arguments given in the reply to justify this are included in the introduction/discussion. [Ok. The aforementioned arguments have been included in the Introduction, and this Part has been partially modified in order to attend better the concerns] In addition, in some cases (e.g. “Pakayaku access is not free.”), the unclarity remains in the text, and the authors only explain what they mean in the reply. This is of course not useful for other readers (who may also wonder what is meant with “free” in this context). I recommend to edit the text where unclarities were previously indicated, if not yet done so. [Yes, these explanations improve the previous draft, so we have included now, as you suggest ] The grammar also still needs to be revised.  [We have sent it to the MPDI Services, and we agree if the editor includes grammatical modifications]. Lastly, I have a few minor suggestions left: (i) remove “perfect” in the first line of the abstract (this relates again to my previous comment on small-scale versus large-scale; on large scales such systems are unlikely sustainable),[ Ok. Removed ] (ii) “primary forest” should be “secondary forest in figure 1, ,[ Sorry, but I cannot find these words, but tropical forest ]  (iii) Figure 6 and 7 (and maybe all figures except figure 1 and 2) are in fact tables, and should thus be named "Table 1”, etc .

This manuscript is a resubmission of an earlier submission. The following is a list of the peer review reports and author responses from that submission.

Round 1

Round 2

Reviewer 1 Report

The authors addresed well my comments and I recommend accepting the paper. However, English language should be carefully reviewed by an English native speaker.

